# Lactoferrin-Derived Peptides as a Control Strategy against Skinborne Staphylococcal Biofilms

**DOI:** 10.3390/biomedicines8090323

**Published:** 2020-09-01

**Authors:** Laura Quintieri, Leonardo Caputo, Linda Monaci, Maria Maddalena Cavalluzzi, Nunzio Denora

**Affiliations:** 1Institute of Sciences of Food Production (CNR-ISPA) National Council of Research, Via G. Amendola, 122/O, 70126 Bari, Italy; laura.quintieri@ispa.cnr.it (L.Q.); linda.monaci@ispa.cnr.it (L.M.); 2Department of Pharmacy-Drug Sciences, University of Studies of Bari Aldo Moro, Via E. Orabona, 4, 70126 Bari, Italy; mariamaddalena.cavalluzzi@uniba.it (M.M.C.); nunzio.denora@uniba.it (N.D.)

**Keywords:** skinborne bacteria, lactoferrin, hydrolysate, natural peptides, lactoferricin, lactoferrampin, biofilm formation, biofilm eradication

## Abstract

Coagulase-negative staphylococci (CoNS) widely colonize the human skin and play an active role in host defense. However, these bacteria may cause malodours and increase infection incidence rate in immune-compromised patients and individuals with catheters and implants. CoNS spreading is favored by biofilm formation that also promotes the release of virulence factors and drug resistance. Biofilm control or eradication by antimicrobial peptides (AMPs) represents an attractive strategy which is worth investigating. In this work, bovine lactoferrin (BLF) hydrolysate (HLF) was in vitro evaluated for its antimicrobial and antibiofilm activities against skin-related coagulase negative and positive staphylococci. Despite a minimal inhibitory concentration (MIC) recorded for HLF ranging from 10 to more than 20 mg/mL, a minimal biofilm inhibitory concentration (MIBC) equal to 2.5 mg/mL was found for most target strains. Conversely, MIBC values referred to the individual peptides, LFcinB or LFmpin (herein purified and identified) were significantly lower. Finally, the application of 2.5 mg/mL HLF solution by dipping and spraying on biofilm-attached glass surfaces also caused a high biofilm eradication rate depending on the incubation time, thus attracting interest for future applications in cosmetic formulation for skin care.

## 1. Introduction

Gram-positive *Staphylococcus* spp. are among the dominant bacteria of the skin, which fall into two main groups: coagulase positive pathogens (CoP, *Staphylococcus aureus*, and *Staphylococcus intermedius*) and rarely pathogenic coagulase negative (CoN) strains that include all the other *Staphylococcus* spp. [1,2,3,4].

Unlike CoP staphylococci, CoN strains are generally considered as the “good” residents, because they play an active role in the maturation and homeostasis of cutaneous immunity [2,5,6]. However, these bacteria may also be involved in infections when skin is compromised: they primarily infect immune-compromised patients and individuals with catheters and implants [4]; erythema, atopic dermatitis, corneal infections, and urinary tract infections are the most recurrent diseases [7,8,9]. Besides their role as infectious agents, coagulase-negative staphylococci (CoNS) are also involved in the malodorous production by humans [10]. Malodours principally result from the transformation of N-acyl glutamine and hydroxyalkyl cysteinylglycine, in the axillary sweat, into volatile fatty acids and thioalcohols by staphylococcal enzymes [10,11].

It has also been demonstrated that many types of infections can originate or progress from persistent staphylococci forms well known as biofilms [12]; *Staphylococcus* biofilms produced both by CoP and CoN strains are one of the leading causes of catheter or implant-associated infections [13], as well as more severe diseases due to the onset of antibiotic resistance, persistent inflammation and delays in healing [14]. Staphylococci in biofilm or planktonic state also produce specific volatile organic compounds (VOCs), causal agents of malodorous wounds. Recently, volatilome of wounds has been used to differentiate chronic wounds from undamaged skin and monitor the efficacy of clinical treatments in wound repair [15,16].

Thus, biofilm control and eradication present several major challenges to topical formulations to prevent and alleviate skin diseases, using compounds targeting the bacterial extracellular matrix, regulatory signaling networks, and horizontal genetic transfer; in fact, the hallmark of the bacterial cells in biofilm state is the increased resistance to stresses including environmental factors, immune system response, disinfectants, and antibiotics [17].

In light of these considerations, antibiofilm strategies might represent a mild way to counteract skin persistence of staphylococci and reduce the microbial antibiotic resistance risk without inducing changes in the biodiversity of skin microbiota and consequently in host defenses.

Ongoing antibiofilm strategies for the treatment of staphylococcal skin diseases include low-amperage direct electrical current exposure [18], functionalized hydrogels and nanomaterials [17], plant extracts [19,20], bacterial metabolites [21], enzymes [22], and antimicrobial peptides (AMPs [23]). These latter are components of innate immunity system showing several advantages: (a) the broad spectrum of action against Gram-positive and negative bacteria, (b) a lower acquired resistance compared to antibiotics, (c) the synergistic interactions with other antimicrobials [23,24].

Due to their chemical and physical properties [25], AMPs have gained increasing consideration, moving current researches towards the discovery of novel active peptide sequences. To this purpose, several biotechnological protocols have been recently developed to obtain active sequences for multiple applications (pharmaceutical field, food preservation, crop protection; [26,27,28,29]. However, to the best of our knowledge most of the studies concerning the AMPs activities in the skin diseases caused by CoNS have referred to endogenous AMPs [23].

Recently, Quintieri et al. [30,31] reported that sub-lethal concentrations of the bovine lactoferrin (BLF) hydrolysate (HLF) by pepsin digestion, mostly containing the AMPs lactoferricin B (LFcinB), had effectively prevented biofilm formation by food-borne antibiotic resistant pseudomonads; these species are retrieved among skin microbiota [32] and the significant impact on their metabolic pathways and regulators was also recorded [30,31]. Peptides were obtained by enzymatic hydrolysis of BLF [33], an iron binding glycoprotein produced by human and animal exocrine glands (tears, saliva, vaginal secretions, and most recently sweat; [34]) and playing an important role in the innate and adaptive immune responses; the obtained results also suggest that BLF enables skin wound healing [35]. Nonetheless, to date there is a lack of thorough studies reporting on the activity of these peptides against skin borne bacteria.

In light of these data, the herein presented work aims at assessing the antimicrobial and antibiofilm activity of HLF against several skin-associated *Staphylococcus* species for its potential future exploitation in biomedicine aiming at the development of clinical strategies for the care of skin diseases. Purified antimicrobial peptides were also identified and assayed for the antibiofilm activity at their sub-lethal concentrations.

## 2. Experimental Section

### 2.1. Bacteria and Culture Conditions

Target CoN strains *Staphylococcus caprae* DSM 20608, *S. epidermidis* (FM6-1, FM96, S71, UR63), *Staphylococcus equorum* subsp. *equorum* DSM 20674, *S. haemolyticus* DSM 20263, *S. saprophyticus* (S15, S17, UR18), *S. xylosus* DSM 20266T, and CoP *S. aureus* LMG 22525 were obtained from the ISPA-CNR microbial collection (Institute of Sciences of Food Production, National Council of Research, Bari, Italy) stored at −80 °C. Before their use, all strains were freshly cultured overnight under aerobic conditions in Tryptic Soy Broth (TSB; Oxoid, Milan, Italy), at 37 °C with continuous agitation (13.61 rad/s). Then, they were refreshed in culture media to reach the optical density (OD) at 600 nm of ca. 0.16 (corresponding to 7.00 log CFU/mL), used in the subsequent experiments as initial inoculum.

### 2.2. Static Biofilm Formation

Biofilm formation was assayed in 96-well microtiter plates (Corning^®^, Corning, NY, USA) and quantified as previously described [36]. Briefly, overnight cultures of each strain were diluted 1:100 into fresh TSB (100 μL) supplemented with 1% glucose (*w*/*v*; TSBG) to reach an initial concentration of 6–7 log CFU/mL; then, samples were incubated at 37 °C for 48 h. Non-inoculated TSBG was used as negative control. At 6, 24, 36, and 48 h, planktonic cell growth was determined by measuring OD_λ = 600 nm_ with a microplate reader (Varioskan Flash, Thermo Fisher, Milan, Italy); in addition, microbial counts were enumerated on TSB agar (TSB amended with 16 g/L of technical agar: TSA) plated with serial 10-fold dilutions. Then, planktonic cells were carefully removed and wells were washed twice with distilled water; biofilm cells adhering to the bottom and side of each well were stained with crystal violet (CV; 0.1%, *w*/*v*). After a second washing step, CV incorporated by biofilm was solubilized with 30% acetic acid (*v*/*v*) and its absorbance (OD) was measured at 570 nm.

### 2.3. Antimicrobial Activity of HLF

The antimicrobial assays were carried out using hydrolysate of bovine lactoferrin (HLF) solutions, obtained by hydrolysis of BLF (NZMP lactoferrin 7100, Fonterra, Boulogne-Billancourt, France) with pepsin according to Quintieri et al. [33]. Then, overnight cultures of *Staphylococcus* spp. strains exhibiting biofilm biomass higher than 0.40 (as CV Abs _λ=570nm_ and at least one strain for specie) were inoculated (ca. 6 log CFU/mL; in triplicate), in sterile Falcon(R) 48-wells polystyrene microplates (BD Biosciences, Erembodegem, Belgium), previously filled with 2 mL of TSBG (control) or TSBG with increasing concentration of HLF (0.625, 1.25, 2.5, 5, and 10 mg/mL). Microplates were incubated at 37 °C for 48 h and microbial counts were determined on TSA at 24, and 48 h. Minimal inhibitory concentration (MIC) values were determined as the lowest concentration which prevented bacterial growth after incubating 48 h at 37 °C [37]. At the end of experiment (48 h), 30 μL of each sample (controls and treated samples) were inoculated in 3 mL of culture media and incubated at 37 °C for 24 h. Then, OD_λ_ = 600 nm was registered; serial 10-fold dilutions of sample in physiological saline plated on TSA were also performed for samples not showing growth. The minimum bactericidal concentration (MBC) was defined as the lowest concentration of antimicrobial agent needed to kill 99.9% of the final inoculum after incubation for 24 h under a standardized set of conditions [37].

### 2.4. LC/MS/MS Analysis of HLF

The HLF was characterized for peptide composition by using the high-performance liquid chromatography with the tandem mass spectrometric (LC/MS/MS) detection method. Briefly, 20 µL of each antimicrobial BLF-derived hydrolysate, was injected in an (U)HPLC pump equipped with an autosampler (AccelaTM, ThermoFisher Scientific, San Jose, CA, USA). The chromatographic separation was accomplished by gradient elution on a reversed phase column Accucore RP-MS (100 × 2.1 mm; 2.6 µm; Thermo Scientific, Waltham, US) at a flow rate of 250 µL/min. The gradient was as follows: from 90% to 65% of solvent A (A = H_2_O + 0.1% of formic acid and reserve B = acetonitrile + 0.1% of formic acid) in 23 min, then isocratic for 2 min, down to 50% in 5 min, kept stable for 3 min, again down to 5% in 40 min and back to 95% in 5 min. This composition was maintained for 14 min to assure column reconditioning. MS analyses were performed to identify the active peptides using a Linear Ion Trap Mass Spectrometer (ThermoFisher Scientific) with an ESI interface (ESI-LTQ Velos Pro). The mass spectrometer was operated in positive ion mode using the data dependent acquisition mode (full ion MS and full ion fragmentation MS/MS alternating events) and the collision induced fragmentation mode (CID). The optimized parameters for the detection were the following: CID fragmentation with a collision energy: 35 V, mass scan range: 250–2000 m/z, minimum signal threshold counts 500, total intensity threshold 200, minimum peak count 8, number of the most intense ions monitored 15, isolation mass width 2 Da; mass accuracy was lower than 0.5 ppm by implementing the mass correction. Peptide identification was carried out by using the Sequest function of the Proteome Discoverer 1.3 software (Thermo Fisher Scientific). The following settings were applied as filters: precursor mass tolerance: 2 Da; fragment mass tolerance: 0.5 Da; peptide confidence medium (False discovery rate 0.05%), unspecific cleavage (no enzyme), variable modification: methionine oxidation. Database searching was carried out by screening an internal small size database (containing allergenic food proteins and other interfering and possible contaminant proteins). Post-searching was filtered by applying the following conditions: minimal peptide length: 6-amino acids; minimal peptide number for protein analyzed: 4; peptide mass tolerance: lower than 200 ppm.

### 2.5. Purification of Antimicrobial Peptides (AMPs) by Gel Filtration Chromatography (GFC) and Their Antimicrobial Activity

The HLF was fractionated by Gel Filtration Chromatography (GFC) on BioSep-SEC-S2000 (ID 300 × 7.8 mm; Phenomenex, Castel Maggiore, BO, Italy) mounted on AktaPurifier 10 system (GE Healthcare, Uppsala, Sweden) equipped with a Pump-900 binary pump, UPC-900multi-wavelength UV-Vis detector and Frac-920 fraction collectors. Freeze-dried samples were dissolved in MilliQ water and injected (2.5 mg) into a column equilibrated with 45% acetonitrile containing 0.1% TFA and eluted under isocratic condition at a flow rate of 1 mL/min. Absorbance peaks revealed at 214 nm, were collected using the automatic peak fractionation function of the UnicornTM software 5.1 release included the HPLC system. A synthetic lactoferrampin (LFmpin; KLLSKAQEKFGKNKSRSFQL, Primm s.r.l., Milan, Italy) was loaded as standard.

A total of 20 runs were performed for the BLF hydrolysate; the fractions with the same retention time were pooled and named GFC fractions. All samples were firstly dried in SpeedVac (SVC 100H; Savant Instruments Inc., Hicksville, NY, USA) and subsequently freeze-dried.

The freeze-dried GFC fractions were re-dissolved in 2 mL of TSB medium (final concentration of 5 mg/mL) and inoculated with an overnight culture of *S. epidermidis* displaying OD _λ = 600nm_ of 0.16 ± 0.05 (corresponding to ca. 7 log CFU/mL), reaching an average final concentration of 3 log CFU/mL. Controls without GFC fractions were also included. Each sample, in triplicate, was incubated at 30 °C for 24 h. Viable counts of each cultures were assessed after 0, 4, 8, and 24 h onto TSA.

### 2.6. Evaluation of Antibiofilm Activity and MBIC Determination

HLF concentrations which did not cause any significant changes in viable cell count (<MIC) were subsequently assayed for the inhibition of biofilm development as reported above. Percentage reductions in biofilm biomass (BBR %) in the presence of different concentrations of HLF were calculated at the different incubation times adopting the following formula:(1)$$\mathrm{BBR}\;(\%)=\left[\frac{{\mathrm{Control}}_{{\mathrm{OD}}_{570\mathrm{nm}}}-{\mathrm{HLFTest}}_{{\mathrm{OD}}_{570\mathrm{nm}}}}{{\mathrm{Control}}_{{\mathrm{OD}}_{570\mathrm{nm}}}}\right]\times 100$$

The minimum biofilm inhibitory concentration (MBIC) was determined as the HLF concentration needed to obtain BBR percentage higher than 50% (HLF-MBIC).

Two antimicrobial peptides, LFcinB [33] and LFmpin, herein identified, were purchased (purity >95%; Gen Script Leiden, The Netherlands) and assayed for antibiofilm against the highest biofilm producers (OD_λ = 570nm_ >2). The assayed concentration were 0.018, 0.037, 0.075, and 0.15 mg/mL for LFcinB, and 0.018, 0.037, 0.075, 0.15, and 0.3 mg/mL for LFmpin. Biofilm was determined as described above after 6, 24 and 48 h of incubation at 37 °C. A mixture of both peptides (0.037 mg/mL) was also included in the experiment. Microbial load in TSBG added or not with LFcinB or LFmpin was also determined at each highest peptide concentration as previously reported. MBIC value was determined as reported above.

### 2.7. Biofilm Eradication by HLF

The ability of HLF to eradicate biofilm from a surface was assessed by two experimental assays against the largest producer of biofilm as reported by Bakkiyaraj et al. and Packiavathy et al. [38,39] with slight modifications. In the first assay, biofilm formed on microscopic slides was eradicated by dipping these latter in an HLF aqueous solution; by contrast in the second one the attached biofilm was eradicated by spray HLF solution.

In both cases, sterile microscope glass cover slips (MS; 10 mm × 10 mm; Agar Scientific Ltd., Stansted, Essex, UK) were transferred in Petri dishes (Ø, 60 mm; Corning, NY, USA) and dipped into 5 mL TSBG. Then, a fresh culture of the selected target strain was inoculated at the concentration of 6–7 log CFU/mL and incubated at 37 °C for 24 h in order to generate the biofilm.

Before the eradication steps, microscope slides were recovered and washed three times in water to removed planktonic cells.

### 2.8. Biofilm Eradication by Dipping in HLF Solutions

Microscope glass cover slides with the attached biofilm formed were transferred to 5 mL of sterile HLF aqueous solutions at different concentrations (2.5, 1.25, and 0.625 mg/mL). Microscope slides dipped in sterile water were included as control. Each sample, performed in triplicate, was incubated at 37 °C for 3 h. After incubation, each microscope slide was recovered, washed twice with sterile water and stained for 15 min with 1 mL of CV (0.1%, *w*/*v*). After two sequential washing steps, biofilm-associated crystal violet was solubilized with 30% acetic acid (1 mL; *v*/*v*) and its optical density was measured at 570 nm.

Percentage (%) of eradicated biofilm was calculated in comparison of water treated slides calculated by following the equation:(2)$$\mathrm{Eradicated}\;\mathrm{biofilm}\;(\%)=\left[\frac{{\mathrm{Control}}_{{\mathrm{OD}}_{570\mathrm{nm}}}-{\mathrm{Test}}_{{\mathrm{OD}}_{570\mathrm{nm}}}\;\mathrm{with}\;\mathrm{HLF}}{{\mathrm{Control}}_{{\mathrm{OD}}_{570\mathrm{nm}}}}\right]\times 100$$

Minimal biofilm-eradication concentration (MBEC) was defined as the lowest concentration of HLF required to eradicate the 50% of biofilm.

### 2.9. Eradication by Sprayed HLF Solution

Each microscope slide with the attached biofilm was transferred to a Petri dish (60 mm; Corning, NY, USA) and sprayed 1, 2 or 3 times with water or with an HLF aqueous solution (2.5 mg/mL). Then, sprayed samples were incubated for 1 h or 3 h at 37 °C. After two washing steps in sterilized water, residual attached biofilm biomass was determined by CV staining as previously reported. The HLF amount sprayed on each surface was determined by recovering volume from 1, 2, or 3 sprays and freeze-drying. Percentage (%) of eradicated biofilm was calculated as reported above.

### 2.10. Statistical Analyses

Statistically significant differences among biofilm biomass values of each assayed staphylococci strain were assessed using one-way ANOVA after checking equality of variances with Levene’s test (*p* < 0.05). Antimicrobial efficacy and biofilm reduction data were analyzed for each assayed strains in relation to different HLF concentrations and time of incubation by using two-way ANOVA. Multi-comparisons were performed by HSD Tukey post hoc test (*p* < 0.05). Whatever requested (no variance homogeneity), a Kruskal-Wallis H test was conducted to evaluate whether during incubation the microbial load of incubation of each assayed strains statistically differed based on increasing HLF concentrations. Subsequently, stepwise step-down comparisons were performed using Dunn’s procedure with a Bonferroni correction for multiple comparisons. Independent Student’s t-test was performed to compare control and GFC-treated samples, as well as biofilm eradication rate for a given spray shot number at different time incubation. Statistical analyses were carried out using the IBM SPSS Statistics (version 20.0, IBM Corp., Armonk, NY, USA) software package.

## 3. Results

### 3.1. Biofilm Formation and Minimal Inhibitory Concentration (MIC) of Lactoferrin Hydrolysate (HLF) against Selected Strains

At 24 h of incubation, according to the results obtained, three distinctive groups were observed: no biofilm producers (OD_λ = 570nm_ < 0.2), moderate biofilm producers (0.4 < OD_λ = 570 nm_ < 1.5), and strong biofilm producers (OD_λ = 570 nm_ >1.5; Figure 1). In general, all assayed strains showed initial biofilm levels (6 h) very low and only for 5 of them (*S. saprophyticus* S15 and UR18, *S. xylosus* DSM 20266, *S. epidermidis* UR63 and FM6–1) were found marked and significant (*p* < 0.05) increases of biofilm amount over incubation time. *S. saprophyticus* UR18 and *S. epidermidis* UR63 reached the highest levels (OD_λ = 570 nm_ ca. 3.833) already at 24 h.

At 6 and 24 h of incubation OD_λ = 600 nm_ of planktonic cells registered values of 0.5 ± 0.1 and 1.05 ± 0.44 (on average), respectively, and corresponding to ca. 8 log cfu/mL; additional 24 h determined an increase that reached an OD_λ = 600 nm_ value of 1.8 ± 0.2 (corresponding to ca. 9 log cfu/mL) only for *S. aureus*.

Based on these results, 6 strains producing moderate (*S. caprae* DSM 20608, and *S. aureus* LMG 22525) and high biofilm amounts (*S. epidermidis* FM6–1 and UR63, *S. saprophyticus* UR18, *S. xylosus* DSM 20266T) were selected to be further investigated for their sensitiveness to HLF.

The antimicrobial activity of HLF for each assayed strain is shown in Appendix A. The heterogeneity of variances did not permit no inferences are permitted on the effects due to main factors (time and HLF concentrations) and their interactions. However, a Kruskal-Wallis H test showed that there was a statistically significant difference in microbial load of each strain over the time between the different HLF treatments (χ^2^(13) = 40.368−37.209, *p* < 0.001). Dunn’s pairwise tests showed a strong evidence (*p* < 0.05, adjusted using the Bonferroni correction) of a difference between the groups time x treatment depending on the strain (Appendix A).

After 24 h of incubation the highest decrease in microbial load (3.55 log cfu/mL, on average) was registered for both *S. epidermidis* strains in presence of HLF equal or higher than 5 mg/mL; by contrast, 5 mg/mL decreased the microbial load of the remaining strains approximately of ca. 1.5 log cfu/mL, on average. No antimicrobial effect was found at lower concentrations. Except for the two most sensitive strains, an additional 24 h of incubation reduced the bacteriostatic effect of 5 mg/mL HLF. At 48 h, the bacteriostatic effect persisted for 3 out of six strains in presence of 10 mg/mL of HLF (Appendix A); this latter concentration also exhibited a bactericidal effect against *S. epidermidis* UR63. Among target strains, only *S. aureus* was the most resistant strain: neither an HLF concentration of 20 mg/mL was able to inhibit its growth.

HLF-MIC and MBC values registered against selected *Staphylococcus* spp. after 48 h of incubation are reported in Table 1.

### 3.2. Identification of Peptides Endowed with Antimicrobial Activity

The identity of all peptides released in HLF is reported in Appendix A. The resulting MS spectrum showed a multi-protonation pattern attributable to a mixture of peptides with molecular masses ranging from 412,2947 to 2096,3055 Da (Appendix A). The multitude of ions detected were attributed to the bovine lactotransferrin protein, upon selecting stringent criteria medium and high confidence level in identification (Appendix A). In addition, to further increase the confidence in peptide/protein identification a threshold of 2 was set as minimum value referred to cross-correlation factor (Xcorr) from Sequest database to enter the list of most reliable peptides identified with the highest confidence. Finally, peptide identification also underwent an internal validation made by visual inspection of the generated MS/MS spectra; only the peptides showing a minimum of three consecutive fragments from the precursor ion were selected as considered the most trustful identification. The MS/MS searching engine (Sequest) used in this work finally allowed to rank sequence candidates according to an assigned score, regardless of the specific scoring system used. As a result, by applying a medium level of stringency, a total of 77 peptides were reliably identified (medium peptide confidence with a false discovery rate of 5%) as displayed in Appendix A. Interestingly, the peptide with amino acid sequence LSKAQEKFGKNKSRSFQL and molecular weight of 2096,3055 Da was detected among the list of identified peptides corresponding to an isoform of peptide LFampin *f*(271–288).

The fractionation of HLF, carried out by GFC, resulted in 5 fractions displaying different elution times (Appendix A). Microbial load of control samples significantly increased from 3.78 to 8.72 log CFU/mL on average from hour 8 of incubation. Only the F3 fraction, corresponding to the retention time of synthetic LFampin *f*(271–288), showed to preserve the antimicrobial activity when assayed against *S. epidermidis*; in particular, F3 caused a remarked reduction of microbial load at 8 and 24 h by an average of 2.57 and 2.92 log cycle, respectively in comparison with the untreated control culture (Appendix A). In light of these findings, LFampin *f*(271–288) and the previously identified LfcinB [33] were assayed in the following experimental trials.

### 3.3. Antibiofilm Activity by HLF and Synthetic Peptides

Basing on previous results, HLF concentrations lower than the MIC value (20 mg/mL for most of strains; Table 1) were assayed for their antibiofilm activity and results reporting the percentages of biofilm reduction by HLF over the time are shown in Figure 2.

Biofilm formation was found to be statistically (*p*< 0.05) affected by HLF at the different concentrations tested, although a specific dose-response relationship was not found at the different incubation times. In accordance with results reported above, *S. aureus* did not produce biofilm before 24 h of incubation; starting from this time of sampling, the lowest HLF concentration of 1.25 mg/mL was enough to reduce by ca. 80% the amount of biofilm biomass. In the interval of time between 24 and 36 h, HLF concentration equal or lower than 1.25 mg/mL also reduced by ca. 75% (on average) *S. epidermidis* UR63 and FM6–1 biofilm biomass; however, additional 12 h registered reduction percentages lower than 50%, on average. At 48 h of incubation a reduction higher than 50 and 90% was instead obtained in the presence of concentrations equal or higher than 2.5 mg/mL.

As concerns *S. xylosus* DSM 20266T and *S. saprophyticus* UR18 biofilms, percentage reduction by ca. 80% was calculated in presence of 2.5 mg/mL starting for 6 h of inhibition; this percentage value increased throughout the incubation period.

Among target strains, the lowest antibiofilm activity by HLF was recorded for DSM 20608. Indeed, after 48 h of incubation only the highest concentration of 10 mg/mL caused the halving of biofilm biomass in comparison to the control sample.Except for this latter strain, MBIC values comprised in the range 1.25 to 2.5 mg/mL for each strain were calculated at 48 h. In addition to HLF, increasing concentrations of the synthetic BLF-derived peptides, LFcinB and LFampin*f*(271–288) were assayed for their antibiofilm activity against the strongest biofilm producers (*S. epidermidis* UR63 and *S. saprophyticus* UR18). No time- or dose-dependent response in biofilm reduction percentage was found for both strains with increasing concentrations of LFcinB and LFampin. However, starting from 6 to reach 48 h of incubation the lowest concentration of LFcinB reduced UR63 biofilm biomass by 31 to 61%, respectively (Figure 3, panel A); this reduction percentage increased when LFcinB was assayed at higher concentrations.

With regard to UR18, 48 h of incubation caused low reduction percentages of the microbial cultures also in presence of the highest peptide concentration (ca. 36 % in presence of 0.15 mg/mL of LFcinB; Figure 3, panel A); at this concentration no differences in the microbial load of planktonic cells were registered between both control cultures and treated ones (on average, 7.6 ± 0.02 and 7.4 ± 0.01 log CFU/mL, respectively).

Similar to LFcinB, each LFmpin concentration proved to effectively reduce UR63 biofilm biomass in a time dependent manner; LFmpin antibiofilm activity against UR18 was higher than LFcinB only at 48 h of incubation (Figure 3, panel B). No antimicrobial effect was recorded at the assayed concentrations as shown by microbial load: 8.2 ± 0.02 and 8.9 ± 0.05 log cfu/mL (on average) in control and treated samples, respectively.

Surprisingly, the mixture of both peptides (0.037 mg/mL) proved to be, on average, more active against both strains compared to the peptides individually assayed. Indeed, the results obtained for *S. epidermidis* UR63 by each peptide were confirmed, whilst the antibiofilm activity against UR18 (Figure 3, panel C) showed to be improved by varying biofilm biomass reduction percentage by 67% already from 24 h of incubation.

### 3.4. Staphylococcal Biofilm Eradication by HLF

The strongest biofilm producer UR63 was selected for the eradication assays and confirmed its ability to produce biofilm also on glass slides within 24 h of incubation (on glass slides CV570nm was 0.71 ± 0.05). Figure 4 shows UR63 residual biofilm biomass on microscope slides after 3-h treatment with increasing concentrations of HLF aqueous solutions. All assayed HLF concentration were able to eradicate biofilm from glass slides. Indeed, the lowest HLF concentrations (0.625 mg/mL) was able to remove biofilm by 56%; thus, this value was also defined as MBEC. After the treatment with HLF at the highest concentration of 2.5 mg/mL, glass slides registered the lowest amount of attached biofilm corresponding to the reduction by 71% in comparison to control samples.

Biofilm eradication by using HLF sprays was also developed. In particular, HLF solution (2.5 mg/mL) was sprayed for a maximum of 3 times on glass slides, subsequently incubated for 1 or 3 h before residual biofilm biomass determination. The eradication rate of biofilm produced by *S. epidermidis* U63 spraying HLF solution (2.5 mg/mL) in relation to number of spray shots and time incubation is shown in Figure 5. The efficacy of the treatment was very low applying only 1 spray (corresponding to 0.54 mg of HLF as dry matter) shot regardless of incubation time; by contrast, 3 spray shots (corresponding to 1.5 mg of HLF as dry matter) led to the highest biofilm reduction percentages (47.60%, on average) without statistically significant difference between 1 and 3 h of incubation. Interestingly, 2 spray shots (corresponding to 0.99 mg of HLF as dry matter) followed by 3 h of incubation showed a significant average increase of 38.12% in biofilm eradication compared to that registered at 1 h (*t*(4)= 7.213, *p* = 0.002).

## 4. Discussion

Antimicrobial peptides (AMPs) are host defense molecules widely studied for their broad spectrum of activity against various gram-positive and negative bacteria, fungi, protozoa, and viruses [40]; mechanisms of action include the perturbation of microbial cell membrane or the modulation of bacterial physiology by binding to DNA [23]. Due to increasing resistance to antibiotics, AMPs are considered as promising approaches leading to novel potential antimicrobial drugs [40].

AMPs are increasingly gaining interest at an industrial level, leading to the development and manufacturing of some products already available on the market both for food and cosmetic uses. In fact, some companies recently addressed a great interest in production of enzymatic whey and casein digests for infant milk formula with several benefits in term of bioactive peptides released by milk proteins [41]. Furthermore, cosmeceutical formulations with biomimetic and bioactive peptides are industrially developed and marketed for stimulating collagen and elastin synthesis and improving skin healing [42]. Furthermore, emerging technologies based on simultaneous in situ enzymatic hydrolysis and fractionation by electrodialysis with ultrafiltration membranes (EDUF) could make the production of AMPs efficient and feasible from generally considered safe proteins [43]. Therefore, although for the first time a lactoferrin hydrolysate has been proposed to control skin-borne staphylococci, the results of our work could be profitably exploited in the production of water-based products compatible with the skin environment.

It has been reported that AMPs, at a concentration lower than their inhibitory concentrations, also exhibit antibiofilm activity due to changes in the metabolic process involved with biofilm formation [44]; they can also interfere with exopolisaccharide components promoting biofilm dispersion [44].

In human skin, bacterial cells in biofilm state may be involved in the etiology, exacerbation, and persistence of chronic wounds and skin disorders [8,12]; thereof, novel AMPs could be investigated to develop control strategies against skinborne *Staphylococcus* spp. without causing alterations to the natural biological barrier [6].

In this perspective, AMPs, with demonstrated antibiofilm activity against *Pseudomonas* spp. [30,31], were herein assayed against CoP and CoN *Staphylococcus* biofilm producers. Among all assayed species, some strains belonging to *S. epidermidis* and *S. saprophyticus* produced the highest biofilm amount, whilst moderate biofilm amounts were recorded for *S. aureus*, *S. caprae*, and *S. xylosus*, *S. epidermidis*, and *S. saprophyticus* species, these being the most occurring species colonizing skin from healthy humans as well as medical implants, urinary catheters, and heart valves [45,46]. Although considered as commensal bacteria, the occurrence of infection might exert a selection pressure favoring biofilm formation with negative effects for human health: infections through venipuncture, atopic dermatitis severity, were, indeed, correlated to community-oriented organisms by all the aforementioned species [3,14,47].

The assayed hydrolysate of bovine lactoferrin(HLF) was previously obtained by digesting bovine lactoferrin with pepsin [33] and was demonstrated to inhibit the growth of gram-negative bacteria in vitro and in food models [33,48,49]. Despite the high initial inoculum, the bacteriostatic effect against the biofilm forming CoN *Staphylococcus* spp. (registered in a strain dependent-manner) was herein reported at HLF concentrations ranging from 5 to 20 mg/mL; a bactericidal effect was also registered for the selected *S. epidermidis* strains at 10 mg/mL. Results agreed with the HLF growth-inhibitory effects previously reported against methicillin-resistant *S. aureus* which registered 8 < MIC values < 64 mg/mL [50]. To the best of our knowledge, no data were instead reported against CoN strains.

As reported by Quintieri et al. [33], the peptide LFcinB was identified as responsible for HLF antimicrobial activity; LFcinB, *f*(17–41; FKCRRWQWRMKKLGAPSITCVRRAF), was released in the hydrolysate at the concentration of 0.043 mg/mg of HLF (as dried weight basis; [33]). In this work, the characterization of peptides with molecular weight lower than 3000 Da released in HLF also allowed to identify the peptide LFmpin *f*(271–288; LSKAQEKFGKNKSRSFQL). Previous studies reported LFmpin isoforms obtained by in silico protocols [51,52,53]; then, synthetic LFampin peptides *f*(265–280), *f*(270–284) and *f*(268–284) were successfully assayed against bacteria and fungi [51,52,53,54]. By contrast, the further LFmpin *f*(271–288) isoform was herein identified in HLF, purified and assayed against a selected strain confirming its antimicrobial activity. Taking into account these results, HLF, LFcinB, *f*(17–41), and LFmpin *f*(271–288) were assayed for their antibiofilm activity.

According to what was previously shown for gram negative bacteria [30,31], HLF negatively affected biofilm production also by CoN strains; interestingly, MBIC value (2.5 mg/mL) was quite similar (3 mg/mL) to that registered for *Pseudomonas* spp. after 48 h of incubation in optimal growth medium [30].

Antibiofilm activity was also maintained by HLF related peptides assayed against the strongest biofilm producers (*S. epidermidis* UR63 and *S. saprophyticus* UR18); a concentration quite similar (0.150 mg/mL) to the LFcinB amount contained in HLF MBIC (ca. 110 µg/mL) was indeed used for both ones. Starting from this value, serial dilutions of peptide solution were also applied. After 48 h of incubation the lowest peptide concentration (0.018 mg/mL), approximately 140 times lower than HLF-MBIC, significantly reduced the biofilm formation by both strains; a similar result was obtained for LFmpin against *S. epidermidis*. By contrast, the reduction of biofilm biomass by *S. saprophyticus* UR18 did not exceed 40% also in the presence of 0.150 mg/mL LFcinB. The subsequent experimental step was addressed to evaluate a putative additive effect by using a mixture of both peptides; interestingly, the treatment with a peptide mixture (0.037 mg/mL) caused a significant percentage reduction in biofilm formation by both strains higher than 60%, starting from 24 h. Data herein reported agreed with a previous work that proved the antibiofilm efficacy of LFcinB in treating systemic or internal infections caused by other bacteria [34]. Moreover, the Lactoferrin chimera (LFchimera), a heterodimeric peptide containing LFmpin *f*(265–284), and a part of LFcinB *f*(17–30), was successfully tested for its antimicrobial and antibiofilm activities against multispecies biofilms derived from subgingival plaque of periodontitis; LFchimera antimicrobial and antibiofilm activities were stronger than chlorhexidine and minocycline hydrochloride, usually used in oral care products [55]. Further evidence concerning the antibiofilm activities were obtained against other microorganisms for human lactoferrin derived peptides, such as LFcinH, *f*(1–11), and its derivatives chemically modified [56,57,58]; lactoferrin, indeed, is widely represented in human and animals, where it exhibits a wide spectrum of biological functions [34]; the amino acid sequences of human lactoferrin and BLF share 69% similarity.

Thus, in accordance with the results of our previous works the evidence reported in this work highlights the promising potential of BLF-derived peptides that, if further investigated, could be exploited to obtain peptidomimetics with improved antibiofilm activity. Recently, in the attempt to overcome problems limiting the general use of AMPs, Svendsen et al. [58] designed LFcinH derived tripeptides with improved drug-like properties; in particular, authors identified the “minimal antibacterial motif” and introduced synthetic amino acid residues improving their antimicrobial and antibiofilm activity. The beneficial effects of this approach also included improved bioavailability, metabolic stability, and rapid mode of action, making them suitable for several applications (i.e., topical use; [58]).

To further support the promising potential of BLF-derived peptides to counteract biofilm formation by skin borne *Staphylococcus* spp., eradication trials were also performed miming putative applications. In particular, *S. epidermidis* biofilm was effectively eradicated from glass surfaces dipped for 3 h in aqueous solutions of HLF at concentration three times lower or equal to its minimal biofilm inhibitory concentration (MBIC) values. *S. epidermidis*, as well as *S. aureus*, are important etiologic agents of microbial ocular infection due to the ability to contaminate and colonize rapidly contact lens, where they grow as biofilm [59]. Likewise, several authors reported the application of antimicrobial and antibiofilm peptides, mainly chemically synthesized, to obtain active coatings preventing microbial growth or biofilm eradicating solutions to be used in prophylaxis of contact lens [60].

Among pharmaceutical technologies, aerosols are widely used in the development of topical drug delivery systems and personal care products [61]; thus, in the attempt to exploit HLF for the development of topical sprays, preliminary results were also obtained in this work. Most of *S. epidermidis* attached biofilm was eradicated by spraying HLF concentration of ca. 1 mg; however, once sprayed, HLF required an incubation period of 3 h to exhibit its activity; putatively, this incubation time allowed the peptide to penetrate the exopolysaccharide matrix promoting biofilm dispersal [44].

In conclusion, we report, for the first time, the antibiofilm activity of BLF derived peptides against skin associated *Staphylococcus* species. Further studies are needed to deepen and validate the efficacy of these peptides or HLF by skin trials aimed at developing appropriate formulations in controlling bacterial biofilm formation also in combination with good practices of skin care.

## Figures and Tables

**Figure 1 biomedicines-08-00323-f001:**
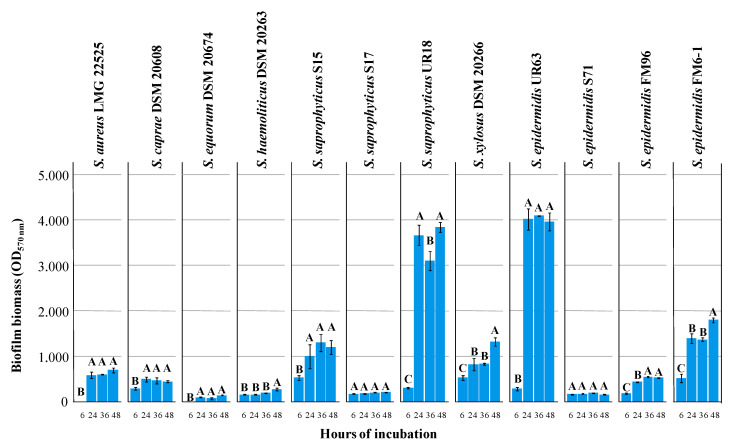
Biofilm biomass produced by *Staphylococcus* spp. strains, grown at 37 °C in TSBG after 6, 24, 36, and 48 h. Values represent the mean ± standard deviation (*n* = 4) and were determined by measuring the absorbance of Crystal Violet (CV) at 570 nm [36]. For each assayed strain values with different letters are significantly (*p* < 0.05) different among days of incubation according to HSD Tukey’s post hoc test.

**Figure 2 biomedicines-08-00323-f002:**
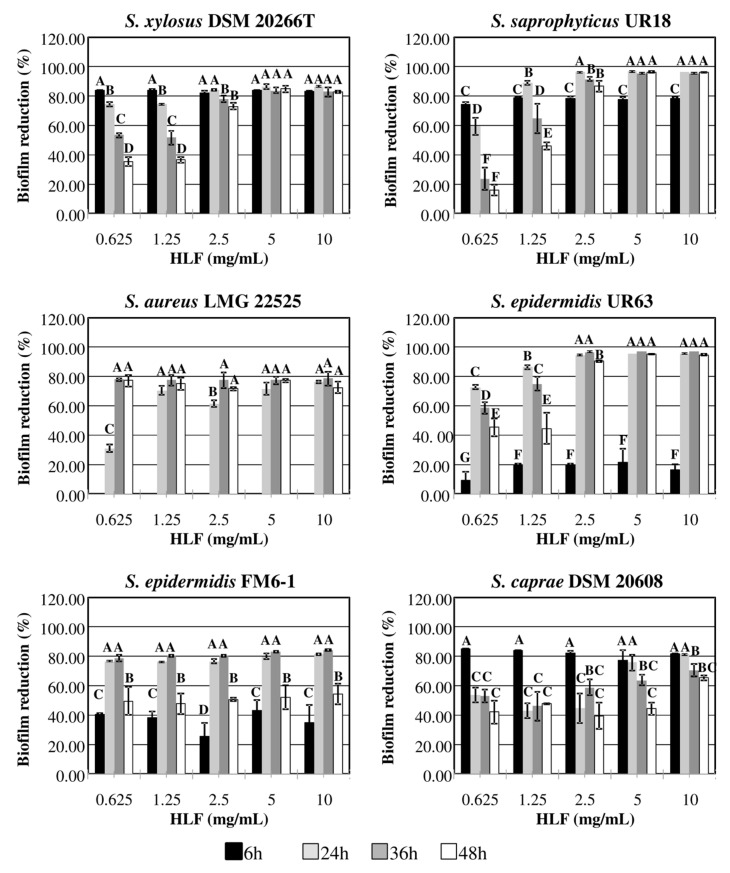
Biofilm reduction percentage (%) by HLF concentrations (0.625, 1.25, 2.5, 5, and 10 mg/mL) added in TSB broth inoculated with *Staphylococcus* spp. at 37 °C and at different incubation times (6, 24, 36, 48 h). The letters above bars show homogeneous subgroups (*p* > 0.05) based on Kruskal-Wallis analysis followed by Dunn’s post hoc tests (*n* = 3). Line represents the calculated minimal biofilm inhibitory concentration (mg/mL; minimal biofilm inhibitory concentration (MIBC)).

**Figure 3 biomedicines-08-00323-f003:**
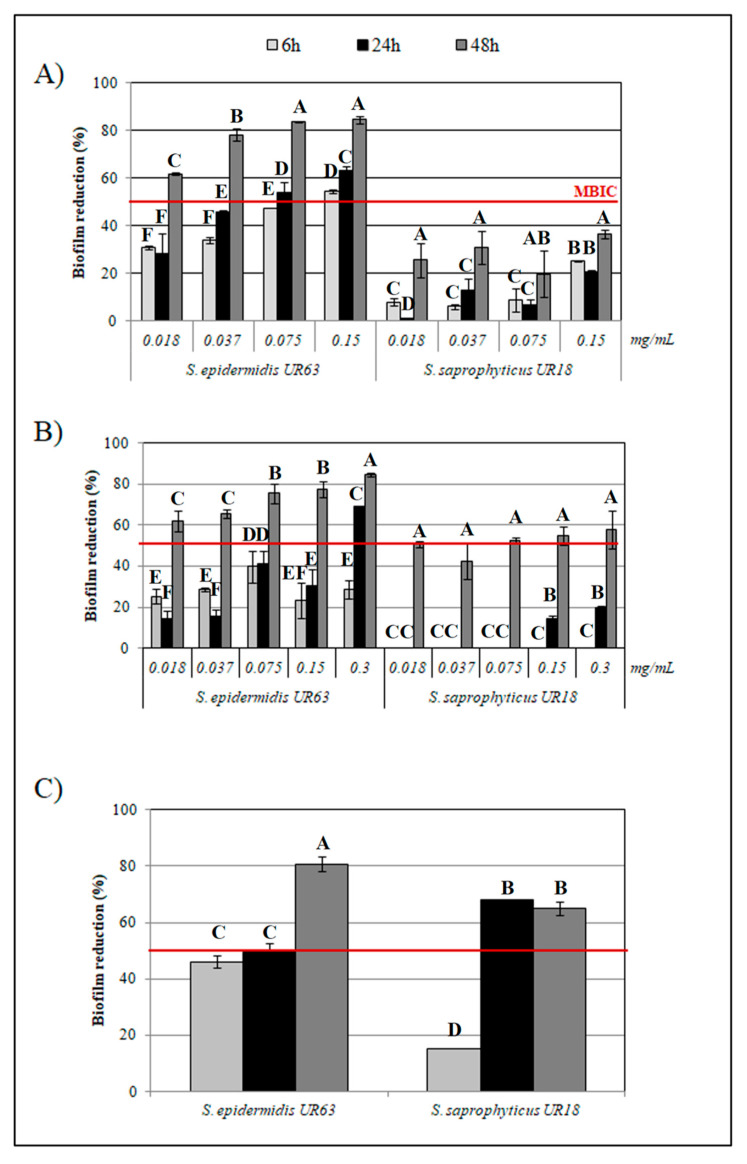
Biofilm reduction percentage in *S. epidermidis* UR63 and *S. saprophyticus* UR18 in vitro cultures amended with (**A**) 0.018, 0.037, 0.075, 0.15 mg/mL of Lactoferricin B (LFcinB); (**B**) 0.018, 0.037, 0.075, 0.15, and 0.3 mg/mL of Lactoferrampin (LFampin); (**C**) LFcinB and LFampin mixture (0.037 mg/mL) at 6, 24, 48 h of incubation. Line represents the calculated minimal biofilm inhibitory concentration (mg/mL; MIBC). Bars are the mean ± standard deviation (*n* = 3). Different letters represent statistically different values (*p* < 0.05) based on Kruskal-Wallis analysis followed by Dunn’s post hoc tests.

**Figure 4 biomedicines-08-00323-f004:**
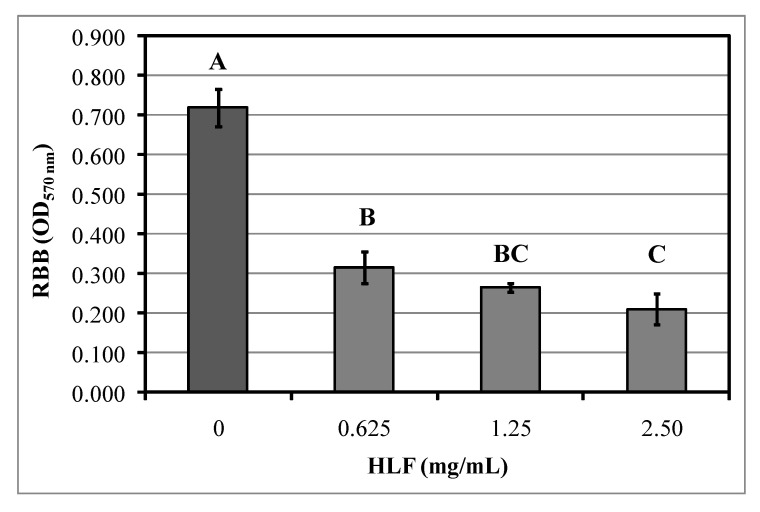
Residual biofilm biomass (RBB) by *S. epidermidis* UR63 determined on microscope cover glass slides dipped for 3 h in bovine lactoferrin hydrolysate (HLF) aqueous solution with increasing concentrations (0, 0.625, 1.25, and 2.5 mg/mL). Bars are mean values ± standard deviations. Different uppercase letters above bars represent statistically different values according Tukey’s test (*p*< 0.05; *n* = 3).

**Figure 5 biomedicines-08-00323-f005:**
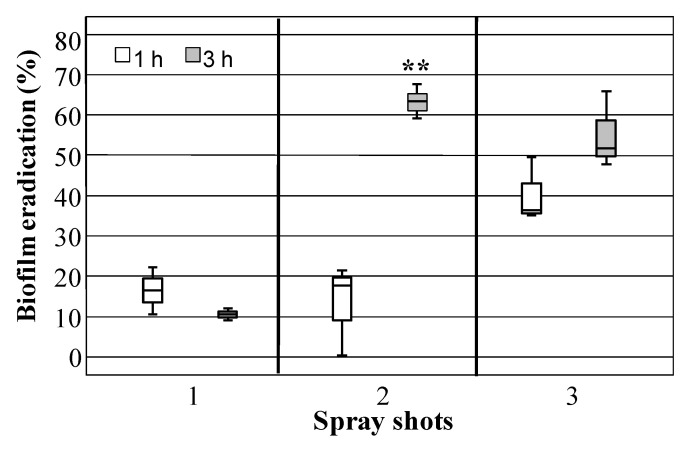
Biofilm eradication percentage of *S. epidermidis* UR63 determined on microscope cover glass slides sprayed one, two, or three times with sterile water or bovine lactoferrin hydrolysate (HLF) aqueous solution (2.5 mg/mL, corresponding to 0.54, 0.99, 1.5 mg of HLF as dry matter) and incubated for 1 or 3 h at 37 °C. ** Asterisks indicate a statistically significant difference (*p* < 0.01) between the incubation time at a fixed spray shot number.

**Table 1 biomedicines-08-00323-t001:** Minimal inhibitory concentration (MIC) and minimum bactericidal concentration (MBC) of bovine lactoferrin hydrolysate (HLF) registered against selected *Staphylococcus* spp. (forming a biomass of biofilm higher than 0.4, as OD _λ = 570 nm_) after 48 h of incubation.

	HLF (mg/mL)	
Strains	MIC	MBC
*S. epidermidis* FM6–1	20	20
*S. epidermidis* UR63	10	10
*S. saprophyticus* UR18	20	>20
*S. xylosus* DSM 20266T	20	>20
*S. aureus* LMG 22525	>20	>20
*S. caprae* DSM 20608	20	20

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
