# Peer review of "Lactoferrin-Derived Peptides as a Control Strategy against Skinborne Staphylococcal Biofilms"

_biomedicines, 2020, doi:10.3390/biomedicines8090323_

Round 1

Reviewer 1 Report

The article by Leonardo Caputo and the team describes the use of peptides derived from bovine lactoferrin pepsin hydrolysate for antibiofilm activity against skin borne staphylococcal biofilms. It is an interesting study that evaluates antimicrobial activity, biofilm eradication of HLF, and synthetic peptides. It is clear that with further studies, including safety assessment, shelf life, and efficacy, the said peptides could be implemented for the production of cosmetics.

In the abstract, the abbreviations AMP and MIC are not needed as they are just written once

I suggest you change the first sentence of the Introduction as follows

Gram-positive Staphylococcus spp. are among the dominant bacteria of the skin, which fall into two main groups: coagulase positive pathogens (CoP, Staphylococcus aureus, and Staphylococcus intermedius) and rarely pathogenic coagulase negative (CoN) strains that include all the other Staphylococcus spp.

Page 2, line 47, What is "VOCs"?

Page 2, line 56, the abbreviation “AR” is not repeated in the rest of the text. So you can delete it.

Page 2, line 60, "AMPs" must be defined as this is it's first appearance in the text. Be acknowledged that the "abstract" stand solitary from the rest of the manuscript.

“at 37°C and 130 rpm” is suggested to be changed as “at 37°C with continuous agitation (130 rpm)”. Also, make sure if the journal allows the use of the unit “rpm” as most journals now prefer the unit “G-Force”.

Page 8, line 290 - 297, It seems unnecessary to divide each of the sentences as separate paragraphs.

Lines are not continuously numbered after Page 9

Discussion,

“antibiofilm activity against Pseudomonas spp. [30-31],” Italicize the word “Pseudomonas”.

“in particular, Authors identified the “minimal antibacterial motif”” The word “Authors” does not need to be capitalized

A major concern/challenge in using enzymatic (pepsin) hydrolysate is the repeatability of producing the same peptides. This limits its industrial applications. I suggest that the authors need to discuss this.

Author Response

The article by Leonardo Caputo and the team describes the use of peptides derived from bovine lactoferrin pepsin hydrolysate for antibiofilm activity against skin borne staphylococcal biofilms. It is an interesting study that evaluates antimicrobial activity, biofilm eradication of HLF, and synthetic peptides. It is clear that with further studies, including safety assessment, shelf life, and efficacy, the said peptides could be implemented for the production of cosmetics.

In the abstract, the abbreviations AMP and MIC are not needed as they are just written once

We adjusted the abbreviations as requested.

I suggest you change the first sentence of the Introduction as follows

Gram-positive Staphylococcus spp. are among the dominant bacteria of the skin, which fall into two main groups: coagulase positive pathogens (CoP, Staphylococcus aureus, and Staphylococcus intermedius) and rarely pathogenic coagulase negative (CoN) strains that include all the other Staphylococcus spp.

We modified the sentence accordingly. Please, see L34-37 of the revised manuscript.

Page 2, line 47, What is "VOCs"?

We explained the abbreviation in the text. Please, see L51 of the revised manuscript.

Page 2, line 56, the abbreviation “AR” is not repeated in the rest of the text. So you can delete it.

Done. Please, see L60 of the revised manuscript.

Page 2, line 60, "AMPs" must be defined as this is it's first appearance in the text. Be acknowledged that the "abstract" stand solitary from the rest of the manuscript.

You are right. We corrected it.

“at 37°C and 130 rpm” is suggested to be changed as “at 37°C with continuous agitation (130 rpm)”. Also, make sure if the journal allows the use of the unit “rpm” as most journals now prefer the unit “G-Force”.

We changed the sentence converting rpm (revolution per minutes) to rad/s units. Please, see L93 of the revised manuscript.

Page 8, line 290 - 297, It seems unnecessary to divide each of the sentences as separate paragraphs.

Thank you. We modified.

Reviewer 2 Report

In the manuscript submitted to BIOMEDICINES (903992) authors works on the use of actoferrin‐derived peptides as a control strategy against skinborne staphylococcal biofilms. This reviewer suggest the publication in BIOMEDICINES only after major revision.

Theme is interesting, techniques are the adequated to solve this kind of analysis, but work fails in the validation and application part.

Validation must be performed by using a reference method applied to the same damples.Application is very poor. The validation and application parts are the main reason for the requirement of MAJOR REVISION.

Minor comments:

* Please correct some typos.

* In Experimental, for Instrumentation, Materials and Reagents, or Programs and Databases (as SPSS, Excel, and others) ever, Product (Manufacturer, City, Country), in this order and format. Please correct in some places. In the case of USA products: Product (Manufacturer, City, State, USA).

Other comments:

* The Novelty Statement is well formulated.ar.

* The Highlights describes adequatly the manuscript.

Author Response

Lines are not continuously numbered after Page 9

Sorry, you are right. We corrected it.

Discussion,

“antibiofilm activity against Pseudomonas spp. [30-31],” Italicize the word “Pseudomonas”.

We corrected. Please, see L390 of the revised manuscript.

“in particular, Authors identified the “minimal antibacterial motif”” The word “Authors” does not need to be capitalized

We corrected it. Please, see L444 of the revised manuscript.

A major concern/challenge in using enzymatic (pepsin) hydrolysate is the repeatability of producing the same peptides. This limits its industrial applications. I suggest that the authors need to discuss this.

In our work we interested to obtain and apply a lactoferrin hydrolysate with the antimicrobial peptide lactoferricin [f(271-288; LSKAQEKFGKNKSRSFQL)]. We previously used the pepsin hydrolysate of lactoferrin in numerous experimental trials against an array of microorganisms confirming the presence of LFcin and the high antimicrobial efficacy (Caputo etal., 2015; Baruzzi et al., 2015; Quintieri et al. 2013-2012). Interestingly, in the present work we found, for the first time in pepsin lactoferrin digest, also lactoferrampin [f(271-288; LSKAQEKFGKNKSRSFQL]. Indeed, those peptides are considered more active than the native lactoferrin itself (as previously demonstrated in silico by Van der Kraan et al. 2004 and Conesa et al. 2008) . Several authors (Abdel-Hamid et al., 2020; Angulo-Zamudio et al., 2019; Chen et al., 2013) have recently confirmed the antimicrobial efficacy of pepsin digest of bovine lactoferrin against a lot bacterial targets, especially CoP staphylococci of clinical interest. In addition, these peptides are putatively released due to the selectivity of pepsin that preferentially cleaves at F, Y W and L in position P1 or P1' of protein (Keil, 1992). Please, see L407-409 of the revised manuscript.

With regard to the industrial applications, some companies recently addressed a great interest in production of enzymatic whey and casein digests for infant milk formula with several benefits in term of bioactive peptides released by milk proteins (for reviewing Fleischer, D. M., Venter, C., & Vandenplas, Y. Hydrolyzed formula for every infant?. In: Protein in neonatal and infant nutrition: Recent updates Bathia J., Shamir R., Vandeplas Y. Eds.,Vol. 86, Nestec Ltd, Vevey S. Karger A.G., Basel, Switzerland, 2016 pp. 51-65.)

Furthermore, cosmeceutical formulations with biomimetic and bioactive peptides are industrially developed and marketed for stimulating collagen and elastin synthesis and improving skin healing (Lima, T. N., & Pedriali Moraes, C. A. (2018). Bioactive peptides: applications and relevance for cosmeceuticals. Cosmetics, 5(1), 21). Therefore, although for the first time a lactoferrin hydrolysate has been proposed to control skin-borne stirrups, the results of our work could be profitably exploited in the production of water-based products compatible with the skin environment.

Ultimately, new recent technologies like the simultaneous in situ hydrolysis and fractionation by electrodialysis with ultrafiltration membranes (EDUF) could make efficient and feasible the production of AMPs from generally considered safe proteins (Suwal, S., Rozoy, E., Manenda, M., Doyen, A., & Bazinet, L., 2017. Comparative study of in situ and ex situ enzymatic hydrolysis of milk protein and separation of bioactive peptides in an electromembrane reactor. ACS Sustainable Chemistry & Engineering, 5(6), 5330-5340.)

Please, we enclosed a brief discussion on your valuable suggestion in L372-382 of the revised manuscript.

In the manuscript submitted to BIOMEDICINES (903992) authors works on the use of actoferrin‐derived peptides as a control strategy against skinborne staphylococcal biofilms. This reviewer suggest the publication in BIOMEDICINES only after major revision.

Theme is interesting, techniques are the adequated to solve this kind of analysis, but work fails in the validation and application part.

Validation must be performed by using a reference method applied to the same damples. Application is very poor. The validation and application parts are the main reason for the requirement of MAJOR REVISION.

You are right. With regard to biofilm eradication trials we used the reference methods previously validated and reported by Bakkiyaraj et al., 2017 and Packiavathy et al., 2014 with slight modifications. Therefore, we included the related citation in L180-182 of the revised manuscript.

Future trials will certainly be oriented to develop a product based on HLF by validating on skin models the results obtained and following the regulatory standards and claims required for its possible marketing.

Minor comments:

* Please correct some typos.

* In Experimental, for Instrumentation, Materials and Reagents, or Programs and Databases (as SPSS, Excel, and others) ever, Product (Manufacturer, City, Country), in this order and format. Please correct in some places. In the case of USA products: Product (Manufacturer, City, State, USA).

We made the required corrections

Other comments:

* The Novelty Statement is well formulated.ar.

* The Highlights describes adequatly the manuscript.

Round 2

Reviewer 1 Report

The authors have done an excellent job during the manuscript revision. I suggest authors to represent the supplementary figures (S1, S2, and S3) in a single document with suitable figure captions. It is OK if the figure captions are available online. Also, I was unable to locate in-text indications for Figures (S2 and S3). Carefully go through the manuscript again during the proofing stage to minimize errors.

Author Response

The authors have done an excellent job during the manuscript revision. I suggest authors to represent the supplementary figures (S1, S2, and S3) in a single document with suitable figure captions. It is OK if the figure captions are available online. Also, I was unable to locate in-text indications for Figures (S2 and S3).

I am happy for the reviewer's flattering comments. Thank you very much.

You are right about the supplementary material. Hoping to meet the publisher's indications we built a single supplementary file containing the figures S1, S2, S3 and Table S1 and the related captions.

Please, see also L454-467 of the revised manuscript

Also, I was unable to locate in-text indications for Figures (S2 and S3).

Please, see Figure S2 in L257-262 of the revised manuscript.

Please, see Figure S3 in L269-275 of the revised manuscript.

Carefully go through the manuscript again during the proofing stage to minimize errors. 

We revised further mistakes throughout manuscript

Reviewer 2 Report

After changes, the manuscript submitted to BIOMEDICINES, could be published in its present form

Author Response

(x) Moderate English changes required 

We again revised the manuscript for English mistakes.

After changes, the manuscript submitted to BIOMEDICINES, could be published in its present form

Thank you for your consideration.